# Allelopathic Inhibition and Mechanism of Quercetin on *Microcystis aeruginosa*

**DOI:** 10.3390/plants12091808

**Published:** 2023-04-28

**Authors:** Qianming Zhao, Ruitong Jiang, Yuxin Shi, Anglu Shen, Peimin He, Liu Shao

**Affiliations:** 1College of Marine Ecology and Environment, Shanghai Ocean University, Shanghai 201306, China; 2Shanghai Engineering Research Center of River and Lake Biochain Construction and Resource Utilization, Shanghai 201702, China; 3Marine Scientific Research Institute, Shanghai Ocean University, Shanghai 201306, China; 4Water Environment & Ecology Engineering Research Center of Shanghai Institution of Higher Education, Shanghai 201306, China

**Keywords:** harmful algal bloom, inhibition mechanism, *Microcystis aeruginosa*, quercetin

## Abstract

The utilization of allelochemicals to inhibit algal overgrowth is a promising approach for controlling harmful algal blooms (HABs). Quercetin has been found to have an allelopathic effect on algae. However, its responsive mechanism needs to be better understood. In the present study, the inhibitory effects of different quercetin concentrations on *M. aeruginosa* were evaluated, and the inhibition mechanisms were explored. The results demonstrated that quercetin significantly inhibited *M. aeruginosa* growth, and the inhibitory effect was concentration-dependent. The inhibition rate of 40 mg L^−1^ quercetin on algal density reached 90.79% after 96 h treatment. The concentration of chlorophyll-*a* (chl-*a*) in treatment groups with quercetin concentrations of 10, 20, and 40 mg L^−1^ decreased by 59.74%, 74.77%, and 80.66% at 96 h, respectively. Furthermore, quercetin affects photosynthesis and damages the cell membrane, respiratory system, and enzyme system. All photosynthetic fluorescence parameters, including the maximum photochemical quantum yield (*F*_v_/*F*_m_), the actual photochemical quantum yield (YII), the maximum relative electron transfer rate (rETR_max_), and light use efficiency (α), exhibited a downtrend after exposure. After treatment with 20 mg L^−1^ quercetin, the nucleic acid and protein content in the algal solution increased, and the respiration rate of algae decreased significantly. Additionally, superoxide dismutase (SOD) activities significantly increased as a response to oxidative stress. In comparison, the activities of ribulose 1,5-biphosphate carboxylase (Rubisco) and phosphoenolpyruvate carboxylase (PEPC) decreased significantly. These results revealed that quercetin could inhibit *M. aeruginosa* by affecting its photosynthesis, respiration, cell membrane, and enzymic system. These results are promising for controlling *M. aeruginosa* effectively.

## 1. Introduction

Cyanobacterial bloom is becoming a severe global environmental problem due to aquatic eutrophication [1,2]. Large-scale cyanobacterial blooms have several negative effects on aquatic ecosystems and human health, such as degrading water quality, giving off terrible odors, and releasing algal toxins [3]. In most eutrophic waters, *Microcystis aeruginosa* is dominant in both quantity and frequency [4]. For instance, there were 4, 3, and 3 moderate cyanobacteria blooms in Taihu Lake in 2017, 2019, and 2020 [5]. Approximately 25–75% of the cyanobacterial bloom water can produce toxins, among which microcystin (MC) is the most widely distributed and most harmful [6]. Moreover, in the past 20 years (2001–2019), *Microcystis* has been the dominant species of winter and spring algal blooms in Dianchi Lake [7]. It was widely accepted that *Microcystis aeruginosa* was more sensitive than most *Chlorophyta* species to allelochemical stress [8]. *M. aeruginosa* is a representative species in freshwater cyanobacterial blooms [9], posing a threat to aquatic organisms and humans as a result of producing microcystins [10]. Microcystin can inhibit the activity of protein phosphatase in cells and destroy the homeostasis of protein phosphorylation, resulting in liver injury, primary liver cancer, and other symptoms [11]. Thus, it is crucial to control the growth of *M. aeruginosa* in waters.

In comparison with physical methods [12] or chemical methods [13,14], the utilization of biological treatment is a relatively cost-effective and environmentally friendly approach in cyanobacteria control [15,16,17]. In recent years, considerable research has been conducted using plants’ allelopathy to control *M. aeruginosa* growth [18,19,20]. However, most studies have focused on the impact of plant extracts or dry powders on the growth of *M. aeruginosa* [21,22,23]. Few studies have been carried out to investigate the effects of single allelochemicals on *M. aeruginosa*. In comparison with plant extracts or dry powders, the purified algicides can be easily stored in standby conditions [8]. Furthermore, the responses of algal cells to allelochemicals were much quicker [24]. Single allelochemicals have the advantages of higher efficiency and better specificity [25].

Quercetin (3,3′,4′,5,7-pentahydroxy-flavone) is a natural flavonoid. Quercetin occurs in numerous plants, is a plant-derived metabolite, and has been found to have various biological activities, such as antiviral, antioxidant, and anti-inflammatory effects [26]. For instance, Tang et al. [27] found that quercetin has a specific inhibitory effect on the bacillus *Staphylococcus aureus*. Li et al. [28] revealed that quercetin had a better inhibitory effect on the bloom-forming dinoflagellate *Phaeocystis globosa* compared to the other 19 flavonoids. Currently, quercetin has a good application prospect in controlling algal blooms [29]. However, its potential mechanism of action on algae remains under investigation, and few reports have been conducted. Furthermore, in the early stage, we explored the inhibitory effects of various plant-derived allelochemicals, including quercetin, kaempferol, luteolin, ginkgolic acid, catechin, isorhamnetin, ginkgolide, and bilobalide, on *M. aeruginosa*. Considering its inhibition rate on algae density and photosynthetic fluorescence parameters, it was concluded that quercetin had a better inhibitory activity on algae. Therefore, quercetin was selected to further explore its algal inhibition mechanism. In the current paper, the effect of quercetin on *M. aeruginosa* was investigated by measuring the growth of algal cells, chlorophyll-*a*, photosynthetic fluorescence parameters, respiration rate, cell membrane permeability, and activities of the antioxidant enzymes Rubisco, PEPC, and ATPase. The aim of the present paper is to explore the inhibitory effect, further understand the algal-inhibitory properties of quercetin on *M. aeruginosa*, and provide reference ideas for the biological control of *M. aeruginosa*.

## 2. Results

### 2.1. Effects of Quercetin on Growth of M. aeruginosa

The effects of different quercetin concentrations on the growth of *M. aeruginosa* are displayed in Figure 1. As a whole, cell abundance of algae under quercetin decreased dramatically and showed dose-dependent damage. During the experimental period, the growing state of *M. aeruginosa* was uneven when exposed to various treatment conditions. As depicted in Figure 1, the cell density of *M. aeruginosa* in the control group (0 mg L^−1^) exhibited a rapid upward trend. In the treatment groups, the inhibition rates increased obviously with increased cultivation time and quercetin concentration. Lower quercetin concentrations (2.5 mg L^−1^ and 5 mg L^−1^) slightly inhibited the growth of *M. aeruginosa* at 48 h, and the inhibition rates were 3.05% and 3.49%, respectively. Meanwhile, higher quercetin concentrations (10, 20, and 40 mg L^−1^) had highly significant inhibition effects on the growth of *M. aeruginosa* at 48 h (*p* < 0.01), and the inhibitory rates enhanced over time. At 96 h, the inhibitory rates reached 55.79%, 68.02%, and 90.79%, respectively. Statistical analysis revealed that the EC50 of quercetin on *M. aeruginosa* at 48 and 96 h was 20.99 mg L^−1^ and 8.76 mg L^−1^, respectively. However, the algae density was first inhibited and then rose again at 96 h with 20 mg L^−1^ treatment. Does this show a reduction in force or an improvement in algal adaptability? This deserves attention. In the treatment group with 40 mg L^−1^ quercetin, the algal density showed a continuous downward trend, all lower than the initial algal density (Figure 1).

### 2.2. Effects of Quercetin on Chl-a Content of M. aeruginosa

Chlorophyll-*a* is an essential component of chlorophyll, and the trend of chl-*a* is always in accord with the algae growth. The effects of different quercetin concentrations on chl-*a* content at 96 h are illustrated in Figure 2. Generally, the chl-*a* content of *M. aeruginosa* decreased as the quercetin concentration increased. After being treated with 2.5 and 5 mg L^−1^ quercetin, chl-*a* contents were 287.14 and 287.04 mg m^−3^, respectively, and inhibition rates reached 20.91% and 20.94%. When the quercetin dose increased to 10, 20, and 40 mg L^−1^, the chl-*a* content of *M. aeruginosa* decreased by 59.74%, 74.77%, and 80.66%, respectively, in comparison with the control group (*p* < 0.01).

### 2.3. Effects of Quercetin on Photosynthesis of M. aeruginosa

Figure 3 depicts the effects of different quercetin concentrations on the photosynthesis of *M. aeruginosa*. Photosynthesis varied based on quercetin concentrations and incubation time. Quercetin at 2.5 mg L^−1^ had no significant effect on *F*_v_/*F*_m_, YII, α, and rETR_max_ during the experimental period. However, a substantial destructive effect on photosynthesis was noticed when the concentration was higher than 5 mg L^−1^. Especially in higher-concentration groups (20 and 40 mg L^−1^), highly significant inhibitory effects were observed at 48 and 96 h (*p* < 0.01). At 48 h, the inhibition rates on *F*_v_/*F*_m_, YII, α, and rETR_max_ reached 98.28%, 98.75%, 96.65%, and 99.64%, respectively, after being exposed to 40 mg L^−1^ quercetin. However, a similar phenomenon as algal density rose deserves attention. In comparison with 48 h, photosynthetic fluorescence parameters were slightly increased at 96 h. Furthermore, the inhibition of *F*_v_/*F*_m_, YII, α, and rETR_max_ fell back to 95.81%, 96.11%, 96.13%, and 97.74%, respectively. Does this show the degradation of quercetin or the improvement of algal adaptability? The phenomenon is worth further studying.

### 2.4. Damage of Quercetin to Cell Membrane of M. aeruginosa

Based on the above results, in order to identify the subsequent mechanism by which quercetin affects *M. aeruginosa*, we decided to use 20 mg L^−1^ quercetin to conduct the subsequent experiment. The influence of quercetin (20 mg L^−1^) on the cell membrane of *M. aeruginosa* was studied by measuring the protein content and nucleic acid in the culture solution at 48 h and 96 h (Figure 4). For the control group, the content of protein and nucleic acid remained stable during the experiment. Their absorbance was always maintained at approximately 0.1. After 48 h exposure, the OD_260_ and OD_280_ of the treatment group increased to 0.76 and 0.80, respectively, and the values decreased slightly to 0.59 and 0.70 at 96 h, respectively. As depicted in Figure 4, the OD_260_ and OD_280_ at both 48 and 96 h indicated a very significant difference from the control (*p* < 0.01). The results indicated that quercetin exposure could result in the leakage of intracellular substances and change the membrane permeability of *M. aeruginosa*.

### 2.5. Effects of Quercetin on Respiration of M. aeruginosa

The effects of 20 mg L^−1^ quercetin on the respiration of *M. aeruginosa* are illustrated in Figure 5. As depicted in Figure 5, the respiratory rate of the control group increased gradually with time. However, in the treatment groups, the respiratory rate of *M. aeruginosa* showed a downtrend. At 96 h, the respiratory rate decreased to 1.00 μmol L^−1^ min^−1^, and the inhibition efficiency was 53.49%, significantly lower than the control group (*p* < 0.01). The results demonstrated that exposure to quercetin leads to a continuing decline in respiratory rates.

### 2.6. Effects of Quercetin on Enzymes Activities of M. aeruginosa

Enzyme activities of *M. aeruginosa* changed significantly after exposure to 20 mg L^−1^ quercetin (Figure 6). After 96 h exposure, the SOD activity increased rapidly, and the relative SOD activity reached 230.04% (Figure 6A). As presented in Figure 6B, the activity of Rubisco tended to decrease in the treatment group, and the inhibitory effect enhanced gradually. At 48 and 96 h, the Rubisco enzyme activity was only 66.92% and 36.13% of the control group. As for PEPC and Ca^2+^-Mg^2+^ATPase, it can be seen from Figure 6C,D that both increased slightly at 48 h and then dropped significantly after being treated with quercetin for 96 h, which were 39.52% and 44.67% of the control group, respectively. Generally, enzyme systems of *M. aeruginosa* were destroyed after exposure to quercetin.

## 3. Discussion

In the current study, the cell density, chl-*a* content, photosynthetic fluorescence parameters, and respiratory rate of *M. aeruginosa* showed a downward trend with quercetin treatment. Our results demonstrated that quercetin showed a significant allelopathic influence on the algae growth. This was also observed in other algal inhibition experiments of allelochemicals, such as juglone, ferulic acid, and coumarin [30,31]. The IC_50,4d_ of quercetin on *M. aeruginosa* was 8.76 mg L^−1^ when the initial density was 2.0 × 10^6^ cells mL^−1^. Li [28] revealed that quercetin could inhibit the growth and photosynthetic system of *P. globose* with IC_50,5d_ of 0.068 mg L^−1^ when the initial algal density was 1.2 × 10^5^ cells mL^−1^. The inhibitory effects of the same allelochemical on different microalgae are quite different, and their inhibitory effects are species-specific. Huang (2016) stated that quercetin could inhibit the growth of *M. aeruginosa* with IC_50, 5d_ of 1.99 mg L^−1^ when the initial algal density was 5 × 10^5^ cells mL^−1^ [32]. It was lower than the IC_50, 4d_ of 8.76 mg L^−1^ in this study with the initial density of 2.0 × 10^6^ cells mL^−1^. The difference in initial algae density may be a reason for the inhibition diversity. Li et al. [33] also found that the inhibition efficiency of soaking *Carex cinerascens* on *Microcystis* growth decreased as the initial algae density increased. Therefore, the difference in setting initial algae density should be paid special attention in order to explore algae inhibition. Moreover, the results of the present study indicated that as the quercetin concentration increased, the inhibition rate of cell density and chl-*a* increased, showing the apparent dose–effect relationship. Other studies have also found similar results [34,35].

In the current paper, the inhibitory effect of quercetin on the growth of *M. aeruginosa* is shown as algistatic and algicidal according to the concentration. An algistatic effect means that the target algae may recover to growth after removing allelochemicals. On the contrary, an algicidal effect means that the target algae are killed, and no algal re-growth is observed. It has been reported that algistatic or algicidal effects may depend on the exposed concentration [36], which also supports our speculation. Furthermore, different allelochemicals may exhibit different algistatic or algicidal effects to a particular stress. Churro et al. [37] reported that bacillamides act as algistatic agents against eukaryotic algae. However, they act as either algicide or algistatic agents against some cyanobacteria based on the concentrations added. Differentiation between these effects will help determine whether the chemicals must be in constant contact with the algae to prevent further growth (algistatic) or the algae has absorbed enough chemicals to eventually die after a sufficient treatment time.

Numerous flavonoids (e.g., kaempferol, apigenin, and luteolin) have been proven to have allelopathy against *Microcystis* [38,39]. Huang [32] demonstrated that 50 mg L^−1^ 6-hydroxy flavone had the highest inhibition rate of 66.03 ± 1.37% on *M. aeruginosa* (with initial algal density of 1.2 × 10^5^ cells mL^−1^) after 5 days. In the present study, the inhibition rate of 40 mg L^−1^ quercetin on the cell density of *M. aeruginosa* reached 90.79% (with initial algal density of 2.0 × 10^6^ cells mL^−1^) at 96 h. Thus, it may be deduced that the inhibition efficiency of quercetin on *M. aeruginosa* is higher than 6-hydroxy flavone. Both 6-hydroxy flavones and quercetin are flavonoids. The difference between their algal inhibitory effects may be the different number of hydroxyls in different flavonoids. Quercetin contains five hydroxyl groups, while 6-hydroxy flavones have only one. Some researchers utilized quantitative structure–activity relationships in order to investigate the relationship between flavonoid structure and algal inhibitory activity.

Furthermore, the more hydroxyl groups flavonoids have, the stronger the algal inhibitory activity is. When the number of hydroxyl groups is two or more, the position of hydroxyl groups can more easily influence the algal inhibitory activity [32]. These conclusions also support our speculation.

Additionally, one other phenomenon deserves attention. The results of this paper indicated that the inhibitory effect of quercetin (higher than 2.5 mg L^−1^) on photosynthetic fluorescence parameters of *M. aeruginosa* increased with the increase of concentration but decreased with time prolongation. The reason may be linked to the fact that quercetin, as a plant-derived allelochemical, is easily degradable [40]. With time extension, quercetin in water is decomposed under light irradiation, decreasing the concentration of active substances [41]. Consequently, its algal inhibitory effect rebounded at 96 h. This is worth paying attention to for further studies of the phenomenon. The stability of quercetin is influenced by factors such as oxygen concentration, pH, temperature, and metal ions. Some studies have demonstrated that the degradation efficiency of quercetin was up to 50% in a pH 13 solution [42]. It was also proven that it will not remain in the environment for a long time and has good ecological safety. The recovery of algal damage is possible as long as the compound concentration is sufficiently low due to degradation. Applying quercetin can be also considered before cyanobacteria communities turn into massive proliferations, which may prevent the development of blooms.

The present study indicated that photosynthesis is not the only target of quercetin. The results of this study indicated that quercetin also inhibits *M. aeruginosa* by damaging the respiratory system of algal cells, possibly due to the effect of algicide on the carbon metabolism pathway of the algae cells [43,44]. This conclusion can be also confirmed by the result of enzyme activity. Rubisco is a crucial enzyme that determines the carbon assimilation rate in photosynthesis. It is also an essential enzyme in plant photorespiration [45]. PEPC is a crucial enzyme in photosynthesis and participates in amino acid metabolic pathways [46,47]. In the present study, the activities of Rubisco and PEPC enzymes related to the respiratory and photosynthetic systems were also inhibited. The above results prove that quercetin has excellent potential in inhibiting the respiration of *M. aeruginosa*.

Cell membrane integrity is usually one of the major indexes for identifying membrane damage [48,49]. When algae suffer an adverse environment, the permeability of the cell membrane will increase, and the intracellular electrolyte will exude [50]. In this study, the content of nucleic acid and protein in treatment groups increased, probably due to the destruction of the cell membrane, resulting in the leakage of intracellular ions and macromolecules [20,51]. The activity of Ca^2+^-Mg^2+^ATPcase is related to the permeability of the cell membrane [52]. Under the stress of quercetin, the Ca^2+^-Mg^2+^ATPase activity of algal cells increased first and then decreased. This indicated that *M. aeruginosa* cells enhanced ion regulation in order to maintain the intracellular and extracellular osmotic pressure balance under quercetin stress. Moreover, with the gradual adaptation to stress, the ATPase activity also began to decrease [52].

Superoxide dismutase (SOD) is a common enzyme that scavenges oxygen-free radicals in organisms. When cells are stressed, the activities of SOD in cells will increase to reduce the harmful oxygen free radicals [20]. In the current study, SOD activity significantly increased under quercetin stress, indicating that *M. aeruginosa* might have oxidative stress reactions to quercetin stress. The reason may be that algae cells enhance the ability to scavenge different reactive oxygen species [53]. Oxidative damage is also an essential mechanism by which tannins inhibit algal growth [54,55]. The SOD activity stimulated by 110 mg L^–1^ tannic acid on the fourth day of the experiment was consistent with the results of this study. Its activity was higher than the control group, indicating that ROS in algal cells was continuously produced under stress, and that SOD was continuously generated by the antioxidant defense mechanism of algal cells to remove ROS [56]. In summary, quercetin can effectively inhibit the growth of *M. aeruginosa* by targeting physiological and biochemical processes. The effects of kaempferol on the MCs level were explored under the same experimental conditions. Kaempferol is a type of flavonoid compound like quercetin. The results indicated no significant difference in the intracellular and extracellular MCs content between the kaempferol and control groups. Considering the phototaxis and planktonic habits of *M. aeruginosa*, quercetin can be applied to lake blooms by surface spraying. The dosage can be determined based on the water volume at a depth of 15 cm below the surface of the lake [57]. The results provide a scientific basis for better understanding the inhibitory effect of quercetin on *M. aeruginosa* and technical support for the application of quercetin to solve the problem of *M. aeruginosa* blooms.

## 4. Materials and Methods

### 4.1. Algae and Culture Conditions

*Microcystis aeruginosa* (FACHB-905) is a type of toxic cyanobacteria which produces microcystins (MCs) during the growth and metabolism process [58]. It was purchased from the Institute of Hydrobiology, Chinese Academy of Sciences (Wuhan, China). The algae were precultured in sterilized BG11 medium under a light intensity of 140 μmol photons m^−2^ s^−1^ provided by a light incubator with a light–dark period of 12:12 h at 25 ± 1 °C. All the culture flasks were shaken three times daily and then rearranged randomly in order to prevent illumination difference.

### 4.2. Experimental Design

A total of 30 mg chemical quercetin (Shanghai McLean Biochemical Technology Co., Ltd., Shanghai, China) was dissolved in 3.0 mL dimethyl sulfoxide (DMSO, purity > 98%, Shanghai McLean Biochemical Technology Co., Ltd.), which was utilized as a stock solution. Quercetin is widely present in various plants [59,60,61], such as *Gingko* leaves with a quercetin content of 13.95%, *Sophora japonica* Linn with 4.1%, and *Allium cepa* skin with 0.14%. *M. aeruginosa* in the logarithmic growth phase was transferred to 100 mL fresh BG11 medium in 250 mL glass flasks with an initial density of 2.0 × 10^6^ cells mL^−1^. Different volumes of quercetin solutions were added to these cultures, reaching a series of concentrations of 0, 2.5, 5, 10, 20, and 40 mg L^−1^. The control groups were the cultures without quercetin solution (0 mg L^−1^). The final DMSO level in culture media was lower than 0.4% (*v*/*v*), which has been proven safe for cells [23]. All the experiments were conducted in triplicate.

According to the efficacy, proper quercetin concentration should be selected for evaluating how far quercetin can damage the cellular membrane, respiration system, and enzyme system of *M. aeruginosa*.

### 4.3. Algal Density and Chl-a Content Measurement

After exposing different concentrations of quercetin, algal density was counted through the use of a hemocytometer (XB-K-25) every 48 h with an optical microscope (OPTIKA, B-1000PH, Ponteranica, Italy). The relative inhibitory rate (IR) was calculated as shown in Equation (1) [62]:(1)$$\mathrm{IR}=\left(1-N/N_{0}\right)\times 100\%.\;$$

*N* is the algal density (cells mL^−1^) of the treatment group, and *N*_0_ is the algal density (cells mL^−1^) of the control group.

Chlorophyll-*a* concentration treated with various quercetin concentrations was measured using spectrophotometry, as demonstrated by Wang et al. [63] with minor modifications. After 96 h, the chl-*a* content was determined by 90% acetone extraction. First, algae cells were collected via suction filtration with 0.45 μm filter membranes (Whatman, Kent County, British). Then, the filter membranes were ground with 90% acetone. Furthermore, the tubes were centrifuged for 10 min at 6000 r min^−1^ (SIGMA, 3-18K, Lower Saxony, Germany), and the supernatants were collected for measurement at absorbances of 750, 663, 645, and 630 nm. The 90% acetone was utilized as the blank solution. Chl-*a* concentration (mg m^−3^) was calculated as shown in Equation (2):(2)$$\mathrm{Chl}-a\left({\mathrm{mg}\;m}^{-3}\right)=\frac{\left[11.64\times \left(D_{663}-D_{750}\right)-2.16\times \left(D_{645}-D_{750}\right)+0.10\times \left(D_{630}-D_{750}\right)\right]\times V_{1}}{V\cdot \delta }.\;$$

*V* is the filtration volume (L), *D* is the absorbance, *V*_1_ is the constant volume (mL), and *δ* is the colorimetric light path (cm).

### 4.4. Assessment of Photosynthetic Fluorescence Characteristics of M. aeruginosa

As regards the treatment with different quercetin concentrations, the photosynthetic activities of *M. aeruginosa* were measured at ambient temperature every 48 h by a pulse-amplitude-modulated (PAM) fluorescence monitoring system (Phyto-PAM, Walz, Bavaria, Germany). The maximum quantum yield of photosystem II can be calculated as (*F*_m_ − *F*_0_)/*F*_m_ = *F*_v_/*F*_m_ after dark adaptation for 10 min [64], representing the maximum photochemical efficiency, with *F*_0_ being the minimal fluorescence, *F*_m_ the maximal fluorescence, and *F*_v_ the variable fluorescence. The actual photochemical efficiency of PSⅡ in the light (YII) was calculated as YII = (*F*’_m_ − *F*)/*F*’_m_, where *F* is the actual fluorescence intensity at any time and *F*’_m_ is the maximal chlorophyll fluorescence intensity of light-acclimated algal suspensions. The other photosynthetic activity parameters (α and rETR_max_) were also tested every 48 h. α indicates algae’s solar energy utilization efficiency, and rETR_max_ presents the maximum relative electron transport rate. 

### 4.5. Determination of Cellular Membrane Damage

The release of intracellular components is a good indicator of membrane integrity. The present paper estimated the release of DNA and RNA from the cytoplasm with an absorbance of 260 nm [65]. The absorbance at 280 nm estimated the protein amount released from the algal cells [66]. After treatment with 20 mg L^−1^ quercetin, 10 mL of *M. aeruginosa* was centrifuged for 10 min at 5000 r min^−1^ in order to remove the algal cells at 0, 48, and 96 h. The optical density (O.D.) of the supernatant was measured at 260 and 280 nm using a UV-Vis spectrophotometer (Shimadzu, UV-1800, Kyoto, Japan). The O.D. values of the experimental and control groups were utilized for assessing the releasing level of nucleotides and protein.

### 4.6. Determination of Respiration Rate of Algal Cells

An aliquot of 2 mL algae was collected separately from the control and treatment groups (20 mg L^−1^ quercetin) at 48 and 96 h. The respiration rate of algal cells in a dark environment was measured using Liquid–Phase Oxygen Electrode System (Chlorolab2+, Hansatech Instruments, Norfolk, British). The details of the method were demonstrated by He et al. [67].

### 4.7. Measurement of Enzymes Activities

An aliquot of 12 mL algae was collected and centrifuged at 10,000 r min^−1^ and 4 °C for 10 min to collect the algal cells. Collected cells were resuspended in 2 mL physiological saline, ground for 5 min in an ice bath with a homogenizer, and then centrifuged at 10,000 r min^−1^ for 5 min [68]. The supernatant was retrieved in order to detect the relative enzyme activities of superoxide dismutase (SOD) and ATPase using the respective specific kits, according to the manufacturer’s instructions (Nanjing Jiancheng Bioengineering Institute, Nanjing, China). The relative activities of ribulose diphosphate carboxylase (Rubisco) and phosphoenolpyruvate carboxylase (PEPC) were determined by enzyme-linked immunosorbent assay (ELISA) (Jiangsu Enzyme Immunological Industry Co., Ltd. Jiangsu, China). The relative enzyme activity was calculated as shown in Equation (3):(3)$$\mathrm{The}\;\mathrm{Relative}\;\mathrm{Activity}\;\left(\%\right)=A_{t}/A_{c}\times 100{\%}_{}.$$

*A_t_* is the enzyme activity of the treatment group, and *A_c_* is the enzyme activity of the control group.

### 4.8. Data Analysis

The data were presented as mean ± SE. For data being normally distributed and with homogeneous variance, statistical differences between the control and treated groups were analyzed using one-way analysis of variance (ANOVA) followed by a Tukey test [63]. Statistical analysis was conducted through the use of SPSS 24 (IBM SPSS Software, Chicago, IL, USA). Figures were generated using Origin 8.0 software (Origin Lab, Northampton, MA, USA). *p*-values of less than 0.05 and 0.01 indicate significant and highly significant differences, respectively.

## 5. Conclusions

Overall, the results of the present study revealed that quercetin could effectively control *M. aeruginosa* growth. The inhibitory effects of different quercetin doses on *M. aeruginosa* growth appear in a dose-dependent manner. Additionally, *M. aeruginosa* was damaged in multiple physiological sites after quercetin exposure, including the photosynthetic system, cell membrane, respiratory system, and enzyme system. These findings illustrate the mechanism of quercetin on *M. aeruginosa* and confirm that quercetin is an excellent allelochemical for controlling the growth of *M. aeruginosa*. However, such a conclusion is still early, and more studies must cover all conditions. Deeper inhibition mechanisms, such as the effect on the production of microcystin and the impact on the gene transcription level of *M. aeruginosa*, should be further studied. Further research will also explore its ecological security by simulating natural conditions in enclosure experiments, which can provide convincing data for future applications in order to suppress cyanobacteria recruitment in aquatic systems.

## Figures and Tables

**Figure 1 plants-12-01808-f001:**
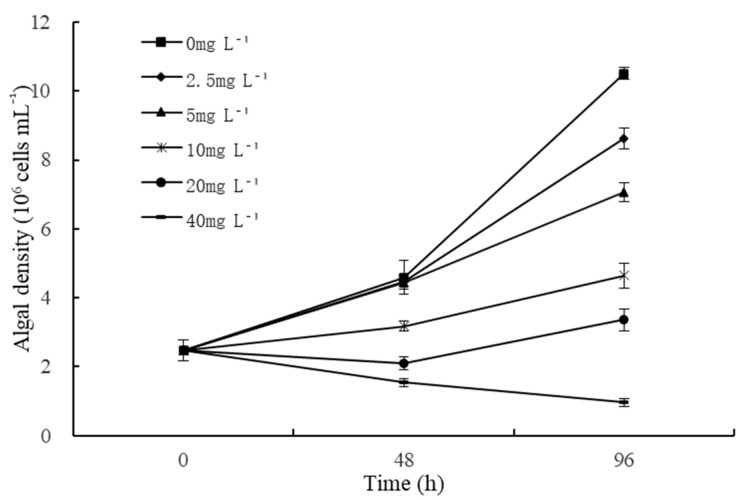
Effects of different quercetin concentrations on algal density of *M. aeruginosa*. The group without quercetin was used as the control.

**Figure 2 plants-12-01808-f002:**
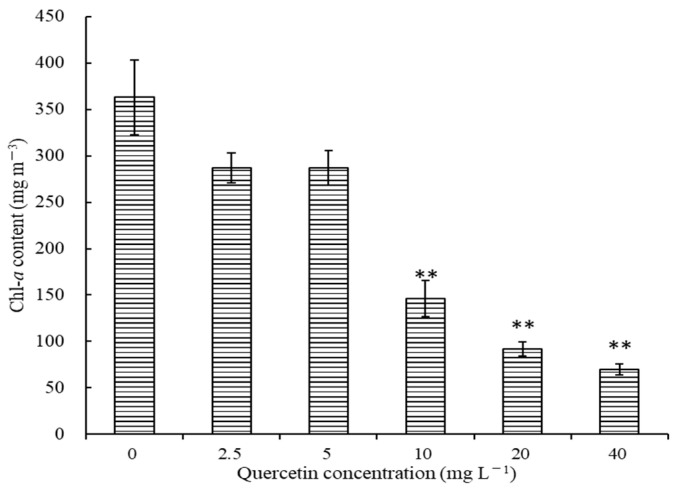
Effects of different quercetin concentrations on chl-*a* of *M. aeruginosa* at 96 h. The group without quercetin was used as the control. ** *p* < 0.01 indicates highly significant differences compared to the corresponding controls.

**Figure 3 plants-12-01808-f003:**
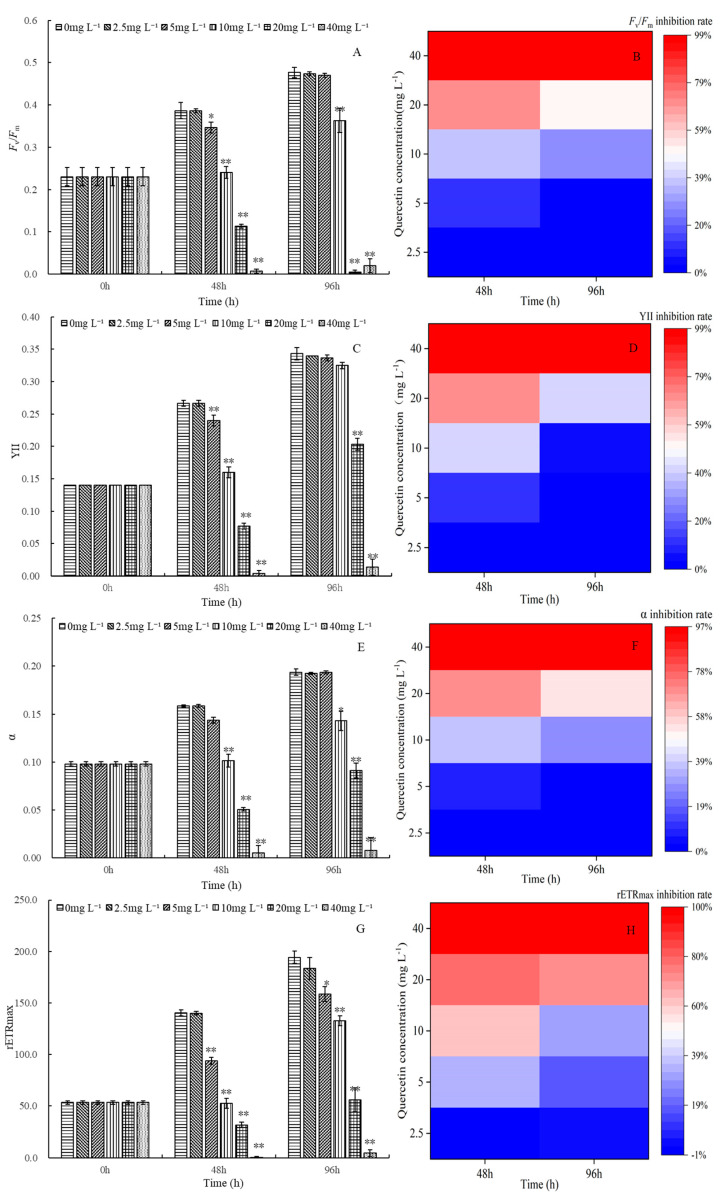
Maximum quantum yield *F_v_*/*F_m_* and its inhibition rate (**A**,**B**), the efficiency of photosystem II YII and its inhibition rate (**C**,**D**), alpha and its inhibition rate (**E**,**F**), rETR_max_ and its inhibition rate (**G**,**H**) of *M. aeruginosa* exposed to different levels of quercetin. The group without quercetin was used as the control. * *p* < 0.05 and ** *p* < 0.01 indicate significant and highly significant differences in comparison with the corresponding controls.

**Figure 4 plants-12-01808-f004:**
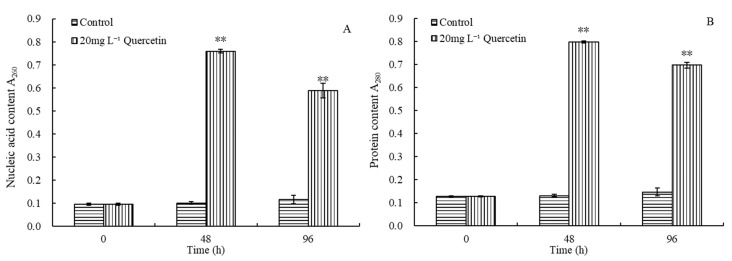
Effects of 20 mg L^−1^ quercetin on nucleic acid (**A**) and protein (**B**) content of *M. aeruginosa*. The group without quercetin was used as the control. ** *p* < 0.01 indicates highly significant differences in comparison with the corresponding controls.

**Figure 5 plants-12-01808-f005:**
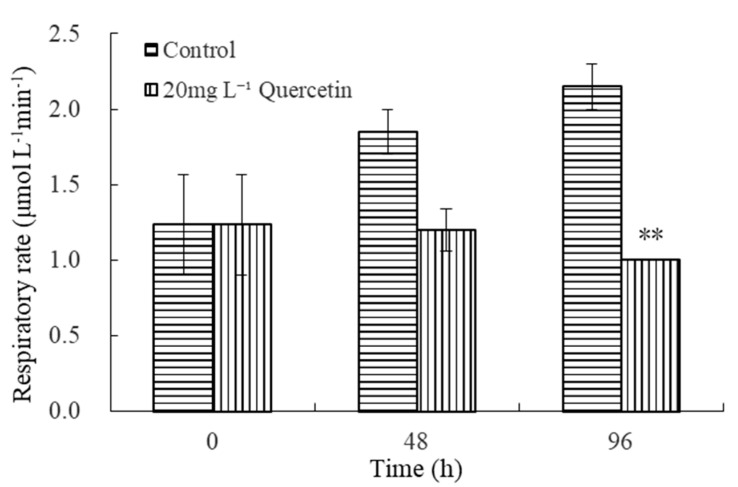
Effects of 20 mg L^−1^ quercetin on respiratory rate of *M. aeruginosa*. The group without quercetin was used as the control. ** *p* < 0.01 indicates highly significant differences in comparison with the corresponding controls.

**Figure 6 plants-12-01808-f006:**
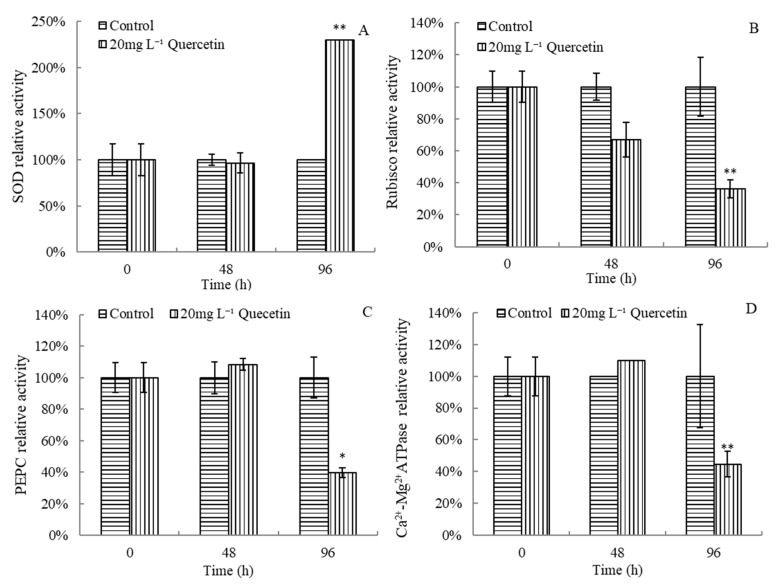
Effects of quercetin on SOD (**A**), Rubisco (**B**), PEPC (**C**), and Ca^2+^-Mg^2+^ATPase (**D**) relative activities of *M. aeruginosa.* * *p* < 0.05 and ** *p* < 0.01 indicate significant and highly significant differences compared with the corresponding controls.

## Data Availability

The datasets generated and/or analyzed during the current study are available from the corresponding author upon reasonable request.

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
