# Peer review of "Allelopathic Inhibition and Mechanism of Quercetin on Microcystis aeruginosa"

_plants, 2023, doi:10.3390/plants12091808_

Round 1

Reviewer 1 Report

I do not have too many problems with following through the paper. Very straightforward and good work.

Author Response

Dear reviewer:

Thanks very much for taking your time to review this manuscript. We really appreciate your encouraging comments and suggestions!

We appreciate for editors/reviewers’ warm work earnestly, and hope the correction will meet with approval. Once again, thank you very much for your comments and suggestions.

Reviewer 2 Report

Dear authors,

the following manuscript titled “Allelopathic inhibition and mechanism of quercetin on Microcystis aeruginosa” has several deficiencies, which I hope you will be able to fix in order to publish your data.

The research presented has more fundamental value and is not practical, because too many questions have no answers. For example: What is the level of the microcystin under quercetin treatment? Is it possible for this treatment to induce the MC levels? How the problem with the fast degradability of the quercetin will be solved? What happens with the quercetin after the 96th hour? What kind of quercetin source you are going to apply for water treatment if you organize it on a large scale? Did you calculate how much quercetin is needed to purify a small water body like a small lake? How you will connect your results with the Plant kingdom? In fact, during your research, you tested one chemical on some growth parameters of one toxic or not toxic (it is not clear from the manuscript) cyanobacteria, which obviously is not a plant.

All this question has to be explained in your revised manuscript. 

Introduction:

Please explain in more detail what is the purpose of your research. Why you are working only with one single component and this component is quercetin? Support your purpose with more literature in order to demonstrate the problem with M. aeruginosa blooms and toxins level detected in China. Show the readers that you are going to solve a local problem. 

Line 54: Please specify what kind of flavonoid is quercetin.

Materials and methods

Line 76: Please explain do you have an idea which source for quercetin (nature/plant origin) you are going to use in the future if you have such a plan? 

Line 85: The price of quercetin is not discussed anywhere in the manuscripts or compared with other components! In fact, the purification of a single component from plant biomass is quite expensive and needs special equipment, etc. Please comment in this sentence only on the quercetin concentration. 

Results:

Line 168: Please explain the observed data.

Line 195:  Please explain the observed data.

Line 207-208: Change the sentence “The influence of quercetin (20 mg L−1) on the cell membrane of M. aeruginosa was studied by measuring the protein content. The nucleic acid in culture solution at 48 h and 96 h (Figure 4)” with "The influence of quercetin (20 mg L−1) on the cell membrane of M. aeruginosa was studied by measuring the protein content and nucleic acid in the culture solution at 48 h and 96 h (Figure 4).

Discussion:

Line 254-255: Please overwrite the sentence. The sentence has no sense.

Line 268-271: Please cite the reference properly. The sentence has no sense. In fact, the cited article (Churro et. al. 2009) is very well written and will be better to read it from the Introduction to the Conclusion in order to improve the quality of your manuscript.

Line 292: How the problem with the fast degradability of the quercetin will be solved?

Line 330-332: How quercetin could be used as "a new algae inhibitor for the purification of eutrophic water" based on your data? 

Author Response

Dear Editors and Reviewers:

Thank you for your letter and for the reviewers’ comments concerning our manuscript. Those comments are all valuable and very helpful for revising and improving our paper, as well as the important guiding significance to our researches. We have studied comments carefully and have made correction which we hope meet with approval. Revised portion are marked in red in the manuscript. The main corrections in the paper and the responds to the reviewer’s comments are as flowing:

Response to the comments from Reviewer #2

The research presented has more fundamental value and is not practical, because too many questions have no answers. For example: What is the level of the microcystin under quercetin treatment? Is it possible for this treatment to induce the MC levels? How the problem with the fast degradability of the quercetin will be solved? What happens with the quercetin after the 96th hour? What kind of quercetin source you are going to apply for water treatment if you organize it on a large scale? Did you calculate how much quercetin is needed to purify a small water body like a small lake? How you will connect your results with the Plant kingdom? In fact, during your research, you tested one chemical on some growth parameters of one toxic or not toxic (it is not clear from the manuscript) cyanobacteria, which obviously is not a plant. All this question has to be explained in your revised manuscript.

Response: Thank you for your comments and the corresponding changes have been marked in red in the revised manuscript.

(1) We explored the effect of kaempferol on the MCs level under the same experimental conditions. Kaempferol is a kind of flavonoid compound that like quercetin. As shown in Figure 1 below, the results showed that there was no significant difference in the intracellular and extracellular MCs content between the kaempferol and control group (P>0.05). Line 369-372, Page12

Figure 1. Effects of kaempferol on the intracellular(a) and extracellular(b) MCs contents of M. aeruginosa.

(2) The stability of quercetin is influenced by factors such as oxygen concentration, pH, temperature, and metal ions. Some studies have found that the degradation efficiency of quercetin was up to 50% in the solution of pH 13 [Lv et al. 2017]. It also proves that it will not remain in the environment for a long time and has good ecological safety. Line 327-334, Page 11

(3) Quercetin is widely present in various plants [Wang et al. 2005; Dong et al. 2016; Gao 2014], such as Gingko leaves with a quercetin content of 13.95%, Sophora japonica Linn with 4.1%, and Allium cepa rind with 0.14%. Line 95-97, Page 2-3

(4) Considering the phototaxis and planktonic habits of M. aeruginosa, quercetin can be applied to lake blooms by surface spraying [Liu et al. 2022]. The dosage can be determined based on the water volume at a depth of 15cm below the surface of the lake. Line 372-375, Page12

(5) Quercetin is a plant-derived metabolite. Line 63-64, Page 2

(6) Microcystis aeruginosa (FACHB-905) is a kind of toxic cyanobacteria, which will produce microcystins (MCs) during the growth and metabolism process [Wang et al. 2022]. Line 85-86, Page2

Comment 1:

Introduction

Please explain in more detail what is the purpose of your research. Why you are working only with one single component and this component is quercetin? Support your purpose with more literature in order to demonstrate the problem with M. aeruginosa blooms and toxins level detected in China. Show the readers that you are going to solve a local problem.

Response: Thank you for the comments and we have added relevant content in the text.

“In most eutrophic waters, Microcystis aeruginosa is dominant in both quantity and frequency [You et al. 2005]. For example, there were 4, 3 and 3 moderate cyanobacteria blooms in the Taihu Lake in 2017, 2019 and 2020 [Zhang et al. 2022]. About 25 % -75 % of the cyanobacterial bloom water can produce toxins, among which Microcystin (MCs) is the most widely distributed and most harmful [Wang Mingliang et al. 2022]. And in the past 20 years (2001-2019), Microcystis has been the dominant species of winter and spring algal blooms in Dianchi Lake [Li et al. 2020].” Line 39-45, Page 1

“Microcystin can inhibit the activity of protein phosphatase in cells and destroy the homeostasis of protein phosphorylation, leading to liver injury, primary liver cancer and other symptoms [Zhang et al. 2017].” Line 48-50, Page2

“This paper aims to explore the inhibitory effect and further understanding the algal inhibitory properties of quercetin on M. aeruginosa and provide reference ideas for the biological control of M. aeruginosa.” Line 80- 82, Page2

“Moreover, in the early stage, we explored the inhibitory effects of various plant derived allelochemicals on M. aeruginosa. Considering its inhibition rate on algae density and photosynthetic fluorescence parameters, it was concluded that quercetin had a better inhibitory activity on algae. Therefore, quercetin was selected to further explore its algae inhibition mechanism.” Line 71-74, Page 2

Comment 2:

Line 54: Please specify what kind of flavonoid is quercetin.

Response: Thank you for the comments and we have added relevant content in the text.

“Quercetin (3,3’,4’,5,7-pentahydroxy-flavone) is a kind of active flavonoid.” Line 63, Page 2

Comment 3:

Materials and methods

Line 76: Please explain do you have an idea which source for quercetin (nature/plant origin) you are going to use in the future if you have such a plan?

Response: Thank you for the comments and We have added relevant content in the text.

“Quercetin is widely present in various plants [Wang et al. 2005; Dong et al. 2016; Gao 2014], such as Gingko leaves with a quercetin content of 13.95%, Sophora japonica Linn with 4.1%, and Allium cepa rind with 0.14%.” Line 95-97, Page 2-3

Comment 4:

Line 85: The price of quercetin is not discussed anywhere in the manuscripts or compared with other components! In fact, the purification of a single component from plant biomass is quite expensive and needs special equipment, etc. Please comment in this sentence only on the quercetin concentration.

Response: Thank you for the comments and the sentence have been revised in the text.

“According to the efficacy, proper quercetin concentration would be picked to assess how far quercetin can damage the cellular membrane, respiration system, and enzyme system of M. aeruginosa.” Line 104, Page3

Comment 5:

Results:

Line 168: Please explain the observed data.

Response: Thank you for the comments and we have added relevant content in the text.

“However, one phenomenon deserves attention, the algae density was first inhibited and then rose again at 96 h in 20 mg L−1 treatment. Does it means the weaken of force or the improvement of algal adaptability. It deserves attention.” Line 188-189, Page4

Comment 6:

Line 195: Please explain the observed data.

Response: Thank you for the comments and we have added relevant content in the text.

“Does it means the degradation of quercetin or the improvement of algal adaptability. The phenomenon is worth further study.” Line 218-229, Page 6

Comment 7:

Line 207-208: Change the sentence “The influence of quercetin (20 mg L1) on the cell membrane of M. aeruginosa was studied by measuring the protein content. The nucleic acid in culture solution at 48 h and 96 h (Figure 4)” with "The influence of quercetin (20 mg L1) on the cell membrane of M. aeruginosa was studied by measuring the protein content and nucleic acid in the culture solution at 48 h and 96 h (Figure 4).

Response: Thank you for the comments and we have revised the sentence in the text. Line 230-232, Page 8

Comment 8:

Discussion:

Line 254-255: Please overwrite the sentence. The sentence has no sense.

Response: Thank you for the comments and the sentence have been revised in the text.

Huang (2016) described that quercetin could inhibit the growth of M. aeruginosa with IC50, 5d 1.99 mg L−1 when the initial algal density is 5×105 cells mL−1. It was lower than the IC50, 4d of 8.76 mg L−1 in this study with the initial density of 2.0 × 106 cells mL−1. The difference in initial algae density may be a reason caused the inhibition diversity.” Line 277-282, Page 10

Comment 9:

Line 268-271: Please cite the reference properly. The sentence has no sense. In fact, the cited article (Churro et. al. 2009) is very well written and will be better to read it from the Introduction to the Conclusion in order to improve the quality of your manuscript.

Response: Thank you for the comments and the sentence have been revised in the text.

“Churro et al. (2009) reported that bacillamides act as algistatic agents against eukaryotic algae but act either as algicide or algistatic agents against some cyanobacteria depending upon the concentrations added. Differentiation between these effects will help determine whether the chemicals must be in constant contact with the algae to prevent further growth (algistatic), or whether the algae has absorbed enough chemicals to eventually die after a sufficient treatment time.” Line 298-303, Page 11

Comment 10:

Line 292: How the problem with the fast degradability of the quercetin will be solved?

Response: Thank you for the comments and we have added relevant content in the text.

The stability of quercetin is influenced by factors such as oxygen concentration, pH, temperature, and metal ions. Some studies have found that the degradation efficiency of quercetin was up to 50% in the solution of pH 13 [Lv et al. 2017]. It also proves that it will not remain in the environment for a long time and has good ecological safety. The recovery of algal damage is possible as long as the concentration of the compound is sufficiently low due to degradation. It can be considered to apply quercetin before cyanobacteria communities turn into massive proliferations, which may prevent the development of blooms.” Line 327-334, Page 11

Comment 11:

Line 330-332: How quercetin could be used as "a new algae inhibitor for the purification of eutrophic water" based on your data?

Response: Thank you for the comments and the sentence have been revised in the text.

The results provide a scientific basis for better understanding the inhibitory effect of quercetin on M. aeruginosa and technical support for the application of quercetin to solve the problem of M. aeruginosa blooms.” Line 377-379, Page12

We appreciate for editors/reviewers’ warm work earnestly, and hope the correction will meet with approval. Once again, thank you very much for your comments and suggestions.

Reviewer 3 Report

Zhao and others presented an interesting manuscript concerning the effects of quercetin in the inhibition Microcystis aeruginosa growth. The design of the experiment was robust, and several physiological aspects were tested, providing new and important insights at the inhibitory properties of quercetin at the target species tested. 

Bellow follows some suggestions that should be addressed for clarity reasons:

a) Concerning figure 3, the authors should increase the legend at the graphic. It gets a little bit confusing to interpret the patterns at the complex graphics.

b) The authors state that quercetin gets degraded over time due to the exposure of sunlight and therefore might explain the observed phenomena on photosynthetic fluorescence parameters of M. aeruginosa.  Considering your results, does the authors think that this phenomenon might not happen if the higher concentration of quercetin (40mg/L) was used?

Author Response

Dear Editors and Reviewers:

Thank you for your letter and for the reviewers’ comments concerning our manuscript. Those comments are all valuable and very helpful for revising and improving our paper, as well as the important guiding significance to our researches. We have studied comments carefully and have made correction which we hope meet with approval. Revised portion are marked in red in the manuscript. The main corrections in the paper and the responds to the reviewer’s comments are as flowing:

Response to the comments from Reviewer #3

Comment 1:

Concerning figure 3, the authors should increase the legend at the graphic. It gets a little bit confusing to interpret the patterns at the complex graphics.

Response: Thank you for your comments. We have now revised the legend of Figure 3 in text as follows. Line220, Page 7

Comment 2:

The authors state that quercetin gets degraded over time due to the exposure of sunlight and therefore might explain the observed phenomena on photosynthetic fluorescence parameters of M. aeruginosa. Considering your results, does the authors think that this phenomenon might not happen if the higher concentration of quercetin (40mg/L) was used?

Response: Thank you for the comments and we have added relevant content in the text. Compared with 20 mg L-1, 40mg L-1 quercetin has a more significant inhibitory effect on the photosynthetic activity of M. aeruginosa. However, at 96h, the inhibition rate of 40mg L-1 quercetin on Fv / fm, YII, α and rETRmax was lower than that at 48h, which was similar to that of 20mg L-1 quercetin.

“The stability of quercetin is influenced by factors such as oxygen concentration, pH, temperature, and metal ions. Some studies have found that the degradation efficiency of quercetin was up to 50% in the solution of pH 13 [Lv et al. 2017]. It also proves that it will not remain in the environment for a long time and has good ecological safety. It can be considered to apply quercetin before cyanobacteria communities turn into massive proliferations, which may prevent the development of blooms.” Line 327-334, Page 11

We appreciate for editors/reviewers’ warm work earnestly, and hope the correction will meet with approval. Once again, thank you very much for your comments and suggestions.

Round 2

Reviewer 2 Report

Dear authors,
Pease see the recomendations and comments for your revised paper in the attached file. There are few moments (lines) where you have to correct or explain. The english also have to be chеckеd again. 

Author Response

Dear Editors and Reviewers:

Thank you for your positive comments and valuable suggestions to improve the quality of our paper. According to your nice suggestions, we have made corresponding modifications and revised portion are marked in red in the manuscript. Also, we apologize for the poor language of our manuscript. We worked on the manuscript for a long time and the repeated addition and removal of sentences and sections obviously led to poor readability. We have now worked on both language and readability and have also involved native English speakers for language corrections. We really hope that the flow and language level have been substantially improved.. The main corrections in the paper and the responds to the reviewer’s comments are as flowing:

Response to the comments from Reviewer #2

Comment 1_12.04.

Line 369-371

I recommend you to delete the results for MCs level under kaemferol treatment. Because the kaempferol is different compound! Finally, in the conclusion you wrote that in the future the activity of mycrocistin coding genes will be assessed under quercetin treatment!

Response: Thank you for your comment and the sentence have been deleted in the text. Line 363-366, Page 12

Comment 2_12.04.

Please, remove from the sentence “a kind of”, it is not a scientific expression!

Response: Thank you for your comment and the sentence have been revised in the text. “Microcystis aeruginosa (FACHB-905) is toxic cyanobacteria” Line 86, Page 2

Comment 3_12.04.

Please, specify which were the different allelochemicals with which you compare the quercetin?

Please remove “etc”. from the end of the sentence - line 78

Response: Thank you for your comment and the sentence have been revised in the text. “Moreover, in the early stage, we explored the inhibitory effects of various plant derived allelochemicals including quercetin, kaempferol, luteolin, ginkgolic acid, catechin, isorhamnetin, ginkgolide, bilobalide on M. aeruginosa.” Line 81, Page 2

Comment 4_12.04.

Please, remove from the sentence “a kind of” active flavonoid. “Kind of” it is not a scientific expression! What mean active flavonoid? You must explain if it’s true?

Response: Thank you for your comment and the sentence have been revised in the text. “Quercetin (3,3’,4’,5,7-pentahydroxy-flavone) is a natural flavonoid.” Line 64, Page 2

Comment 5_12.04.

Line 294: Please, change the word algae with allelochemicals! The sentence “Moreover, different algae may exhibit different algistatic or algicidal effects to a particular stress”, has no sense!

Response: Thank you for your comment and the sentence have been revised in the text. “Moreover, different allelochemicals may exhibit different algistatic or algicidal effects to a particular stress” Line 292-300, Page 11

Comment 6_12.04.

Line 313: Please, change the order of words with “have only one”

Response: Thank you for your comment and the sentence have been revised in the text. “Quercetin contains five hydroxyl groups, while 6-hydroxy flavones have only one.” Line 317, Page 11

Conclusion:

Comment 7_12.04.

Line 387- 389

Please correct the sentences from “However, this research direction is still early, and

studies have yet to cover all conditions. Deeper inhibition mechanisms, such as the

effect on the production of microcystin and the impact on the gene level of M.

aeruginosa, should be further studied”.

to

However, for such conclusion this research direction is still early, and more studies

have yet to cover all conditions. Deeper inhibition mechanisms, such as the effect on

the production of microcystin and the impact on the gene transcription level of M.

aeruginosa, should be further studied.

Response: Thank you for your comments and suggestion concerning our manuscript. The comments and suggestions are all valuable and helpful for revising and improving our paper, as well as the essential guiding significance to our research. We have revised the sentence in the text. Line 391-393, Page 12

We appreciate for editors/reviewers’ warm work earnestly, and hope the correction will meet with approval. Once again, thank you very much for your comments and suggestions.